# Comparison of the Effects of Propofol and Sevoflurane Anesthesia on Optic Nerve Sheath Diameter in Robot-Assisted Laparoscopic Gynecology Surgery: A Randomized Controlled Trial

**DOI:** 10.3390/jcm11082161

**Published:** 2022-04-12

**Authors:** Jung Eun Kim, Seong Yoon Koh, In-Jung Jun

**Affiliations:** 1Department of Anesthesiology and Pain Medicine, Kangnam Sacred Heart Hospital, University of Hallym College of Medicine, Seoul 05355, Korea; geri200@hallym.or.kr (J.E.K.); kohsungyoon@hallym.or.kr (S.Y.K.); 2Department of Anesthesiology and Pain Medicine, Sanggye Paik Hospital, Inje University College of Medicine, Seoul 01757, Korea

**Keywords:** propofol, sevoflurane, optic nerve sheath diameter, robot-assisted laparoscopic gynecology surgery

## Abstract

Optic nerve sheath diameter (ONSD) is used as a surrogate parameter for intracranial pressure. This study was conducted to evaluate the effect of the anesthetics (sevoflurane and propofol) on ONSD in women undergoing robotic surgery. The 42 patients who were scheduled for robot-assisted gynecology surgery were randomly allocated to the sevoflurane group or the propofol group. ONSD was recorded at 10 min after the induction of anesthesia (T0); 5 min, 20 min, and 40 min after carbon dioxide pneumoperitoneum was induced and the patients were put in a steep Trendelenburg position (T1, T2, and T3, respectively); and at skin closure after desufflation of the pneumoperitoneum (T4). Patients were observed for postoperative nausea and vomiting (PONV) during the immediate postoperative period. The propofol group had significantly lower ONSD than the sevoflurane group at T3. Mean ONSD values continuously increased from T0 to T3 in both groups. Two patients in the sevoflurane group experienced PONV. This study suggests that propofol anesthesia caused a lower increase in ONSD than sevoflurane anesthesia.

## 1. Introduction

Robot-assisted laparoscopic surgery has become a popular option for treating gynecological patients because it takes a similar amount of operative time and has lower complication rates and shorter recovery periods than conventional laparoscopic surgery [1]. However, it requires patients to be placed in a steep Trendelenburg position and the induction of carbon dioxide pneumoperitoneum for surgical exposure, both of which increase intracranial pressure (ICP) during robotic surgeries likewise in conventional laparoscopic surgeries [2]. Ultrasonographic measurement of the optic nerve sheath diameter (ONSD) is a reliable method as a surrogate parameter of ICP during the perioperative period [3,4]. Many investigators have expressed concern about the intraoperative measurement of the ONSD with ultrasonography during robotic surgeries [5,6,7,8]. However, studies on various factors that affect the ONSD during robotic surgeries have mainly focused on male patients undergoing robot-assisted laparoscopic prostatectomy.

Anesthetic agents used to maintain anesthesia have variable effects on ICP. Volatile anesthetic agents are potent cerebral vasodilators in anesthetic dosages because they increase cerebral blood volume and ICP in a dose-dependent manner [9]. However, propofol reduces cerebral metabolic rate and cerebral blood flow, thereby reducing ICP [10].

The aim of this study was to evaluate the effect of the anesthetic agents (sevoflurane and propofol) on female patients during robot-assisted laparoscopic surgeries using ultrasonographically measured ONSD as a surrogate metric for ICP. We hypothesized that the ONSD in female patients undergoing robot-assisted surgery increases significantly and that propofol anesthesia produces smaller ONSD increase than sevoflurane anesthesia.

## 2. Material and Methods

### 2.1. Participants

After receiving approval from the Institutional Review Board of Hallym Sacred Heart Hospital (approval number: 2018-09-004) and registering the study on ClinicalTrials.gov (NCT03701529: accessed on 10 October 2018), this prospective randomized controlled trial was conducted. The study included 42 patients between 18 and 65 years of age undergoing robot-assisted laparoscopic hysterectomy or myomectomy between October 2018 and September 2019 at Kangnam Sacred Heart Hospital. Informed consent was obtained from all participants. The exclusion criteria were as follows: history of cerebrovascular incidents, glaucoma or any signs of increased intraocular pressure, liver disease, end-stage renal disease, weight of less than 40 or over 100 kg, and refusal to participate in the study.

### 2.2. Anesthetic and Surgical Techniques

One day prior to surgery, an anesthesiologist used the sealed envelope method to randomly assign patients to either the sevoflurane group or the propofol group. After arriving at the operating room, standard monitoring devices, including electrocardiography, pulse oximetry, noninvasive blood pressure measurement, and entropy measurement devices (GE Healthcare Finland, Helsinki, Finland) were applied to the patients. After preoxygenation, patients in the sevoflurane group received 2 mg/kg of propofol to induce anesthesia, which was maintained with 1.5–2.5% of sevoflurane and 0.05–0.15 mcg/kg/min of remifentanil. Patients in the propofol group received a continuous infusion of propofol and remifentanil using a target-controlled infusion pump. Propofol was titrated within 2–5 mcg/mL, and remifentanil was titrated within 2–5 ng/mL.

Both groups received 0.6 mg/kg of rocuronium and were then intubated. Mechanical ventilation was maintained in a volume-controlled mode with a tidal volume of 6 mL/kg of ideal body weight. The respiratory rate was regulated to maintain an end-tidal carbon dioxide partial pressure of 30–35 mmHg. The radial artery was then cannulated for continuous arterial blood pressure monitoring and was recorded for analysis. Pulse pressure variation measured through arterial blood pressure was recorded in the patient monitor (CARESCAPE Monitor B850, GE Healthcare). All patients were under normothermia and entropy was maintained at 40–60 to ensure that both group of patients were equally well anesthetized. During the operation, carbon dioxide pneumoperitoneum was induced with an intra-abdominal pressure of 10–15 cmH_2_O. Patients were put in a steep Trendelenburg position of 30° and surgery was conducted using a da Vinci robot system (Intuitive Surgical, Inc., Sunnyvale, CA, USA). Surgery was conducted by a highly experienced surgeon.

Patients were removed from the study if they experienced unstable vital signs, had a peak airway pressure ≥ 35 cmH_2_O, had to receive open abdominal surgery, or their ONSD was unable to be measured at any of the predetermined times for any reason.

### 2.3. Measurements

The ONSD was ultrasonographically measured with a 7.5 MHz linear probe by an experienced anesthesiologist (IJJ) who did not know which group the patient was in. The investigator had an extensive experience of ultrasonographic ONSD measurement of more than 100 scans. After 25 scans, a wide range of physicians can reliably measure the ONSD ultrasonographically [11,12]. During ONSD measurement, the anesthetic vaporizer and target-controlled infusion pump were covered with opaque coverings. Standardized criteria were used to optimize ONSD measurement [13]. The linear probe was placed taped to the patients’ closed eyelids with transparent Tegaderm after ultrasound gel was applied. The vitreous body, optic disc, and hypoechoic optic nerve sheath were visualized by gently adjusting the probe angle (Figure 1). By using electronic calipers, the ONSD was measured vertically 3 mm behind the optic disc in the sagittal and transverse planes in both eyes. We used the average of the four values for analysis. Each measurement was completed within one minute.

The ONSD was measured 10 min after the induction of anesthesia (T0); 5 min, 20 min, and 40 min after carbon dioxide pneumoperitoneum was induced and the patient was put in a steep Trendelenburg position (T1, T2, T3, respectively); and at skin closure after desufflation of pneumoperitoneum in the supine position (T4).

Hemodynamic and respiratory parameters, including end-tidal carbon dioxide, blood pressure, heart rate, pulse oximetry oxygen saturation, pulse pressure variation, and peak airway pressure were continuously measured and collected at each of these five points in time. Patients were observed for PONV in the post-anesthesia care unit for one hour.

### 2.4. Statistical Analysis

To calculate the sample size required for this study, it was assumed that the mean difference in the ONSD between the sevoflurane group and the propofol group would be 0.3 mm for the primary end point at 40 min after pneumoperitoneum was induced and the patient was in the Trendelenburg position based on a study by Yu et al. [14]. With an alpha error of 0.05 and a power of 80%, 19 participants per group would be required to detect a difference. To account for dropouts or protocol violations, 21 participants were included in each group.

The normality of the continuous data was analyzed with the Shapiro–Wilk test. Continuous data are presented as mean ± standard deviation or median (interquartile range). Categorical data are presented as frequency (%). Data were compared using an independent two-sample *t*-test, Mann–Whitney U test, chi-square test, or Fisher’s exact test as appropriate. A linear mixed model was used to evaluate the changes in the ONSDs according to time. A *p*-value < 0.05 was considered to be statistically significant. All data were analyzed with SPSS software version 18.0 (IBM Corp., Armonk, NY, USA).

## 3. Results

A total of 45 patients were enrolled in this study. One patient was excluded for meeting the exclusion criterion of weighing over 100 kg, so 44 patients were analyzed and randomly allocated to the sevoflurane group or the propofol group. Two patients were excluded from the analysis because their ONSDs could not be measured. Thus, a total of 42 patients were included in the analysis (Figure 2).

There was no significant difference in the demographic and perioperative data of the two groups (Table 1) or the ONSDs at T0, T1, T2, and T4 (Table 2). At T3, which was 40 min after carbon dioxide pneumoperitoneum was induced and the patient was placed in a steep Trendelenburg position, the ONSDs of the Sevoflurane group were statistically higher than those of the propofol group (*p* = 0.025). The ONSDs of the sevoflurane group increased by 0.64 mm, while those of the propofol group only increased by 0.33 mm. The mean ONSD values of both groups increased significantly at T1, T2, and at T3 compared to T0 (Table 2).

The end-tidal carbon dioxide, mean blood pressure, diastolic blood pressure, and pulse pressure variation values did not differ statistically over time for either group (Table 3). Neither group experienced neurological complications, namely, transient ischemic attack or headache. Two patients from the Sevoflurane group experienced PONV, but this number was not statistically significant.

## 4. Discussion

Compared to diverse studies on ONSD in male patients, there are only few data on ONSD in female patients [15]. Our study has the advantage of investigating the changes in ONSD and the effect of anesthetics on the changes focused in female patients undergoing robot-assisted laparoscopic surgery. We expected significant increase in the ONSD of female patients with a smaller ONSD increase during propofol anesthesia compared to sevoflurane anesthesia.

The results of this study showed that the ONSDs of the propofol group were significantly smaller than those of the Sevoflurane group 40 min after pneumoperitoneum was induced and patients were put in a steep Trendelenburg position. The mean ONSD values of both groups continuously increased during pneumoperitoneum and while the patient was in the Trendelenburg position. This increase in this study of female patients undergoing robot-assisted surgery was consistent with the results of previous studies on male patients undergoing robot-assisted laparoscopic prostatectomy with either inhalation anesthesia or propofol anesthesia; although, the ranges of the increase were different [5,16,17].

Propofol and sevoflurane have different effects on ICP [18,19]. Sevoflurane produces an intrinsic cerebral vasodilatory effect resulting from vascular smooth muscle cell relaxation mediated by calcium and potassium ions [20]. Thus, cerebral blood flow and ICP increase during sevoflurane anesthesia under clinical anesthetic dosages, which are above 1.0 MAC [21]. Propofol causes cerebral vasoconstriction, and cerebral blood flow is reduced following cerebral metabolic rate (CMRO_2_) suppression [22,23]. Cerebral blood flow decreases relative to the dose-dependent depression of CMRO_2_ during propofol anesthesia [23]. Therefore, propofol reduces or maintains ICP. During robot-assisted surgery, pneumoperitoneum displaces the diaphragm, cranially increasing intrathoracic pressure, which increases central venous pressure and ICP [24]. In a Trendelenburg position, gravity exacerbates ICP [25]. Sevoflurane’s cerebral vasodilatory effect acutely increases ICP in patients with increased cerebral blood flow due to pneumoperitoneum and being in a Trendelenburg position [26].

The maximum increase in the mean ONSD was 0.64 mm (14.1%) in the sevoflurane group and 0.33 mm (7.17%) in the propofol group after pneumoperitoneum was induced and the patient was put in a Trendelenburg position. Studies on male patients undergoing laparoscopic surgery have also shown that the ONSD increases more under inhaled anesthesia than under propofol anesthesia [4,14,16,17]. Yu et al. (2019) observed patients undergoing robot-assisted laparoscopic prostatectomy and reported a greater increase in the mean ONSD in patients both under sevoflurane anesthesia (0.83 mm (17.5%)) and propofol anesthesia (0.52 mm (10.9%)) than observed in this study [17]. The average age of patients in their study was 65, which was higher than the average age of patients in this study, which was 44. Older patients have a decreased ability to compensate for changes in intracranial pressure, causing them to have larger ONSDs [6]. Additionally, other clinical factors, such as the degree of the Trendelenburg position, abdominal pressure, end-tidal carbon dioxide, and mean blood pressure, as well as various demographic factors may also have contributed to the difference in the maximum increase in ONSDs as a surrogate of ICP during surgery [6]. To date, the effect of sex on the change in the ONSD has not been investigated. Further large randomized clinical studies are warranted to investigate the effect of sex on the change in the ONSD.

PONV occurs more frequently after robot-assisted surgeries than laparoscopic or open abdominal surgeries [27]. During laparoscopic surgeries in the Trendelenburg position, increased ONSD as a surrogate parameter of ICP is highly correlated with the occurrence of PONV [28]. Other factors, including female and anesthetic technique, increase the risk of PONV [29]. In this study, two patients from the Sevoflurane group experienced PONV even though they were administered antiemetics, but these occurrences were not statistically significant. PONV is closely related to patient satisfaction, length of hospital stay, and morbidity, so female patients undergoing robotic surgery should be given a suitable form of anesthesia [28,30]. Total intravenous anesthesia using propofol produces fewer incidences of PONV than inhaled anesthesia [31,32].

This study had two major limitations. First, the ONSD was monitored until 40 min after pneumoperitoneum was induced and the patient was put in a Trendelenburg position. The ONSD could not be measured after that because the uterus removal interrupted the pneumoperitoneum, which would affect the ONSD. ONSD distension is instantaneous and measuring it is useful for determining whether acute changes have occurred; it was assumed that 40 min was sufficiently long for observing trends in ONSD changes [25]. Second, the sample size was not big enough to design an additional analysis including area under the curve analysis of the ONSD. Further large randomized studies are necessary.

## 5. Conclusions

The ONSD increased significantly in the sevoflurane group compared to the propofol group at 40 min with pneumoperitoneum in a Trendelenburg position. Propofol anesthesia produced smaller ONSD increases than sevoflurane anesthesia in women undergoing robot-assisted surgery.

## Figures and Tables

**Figure 1 jcm-11-02161-f001:**
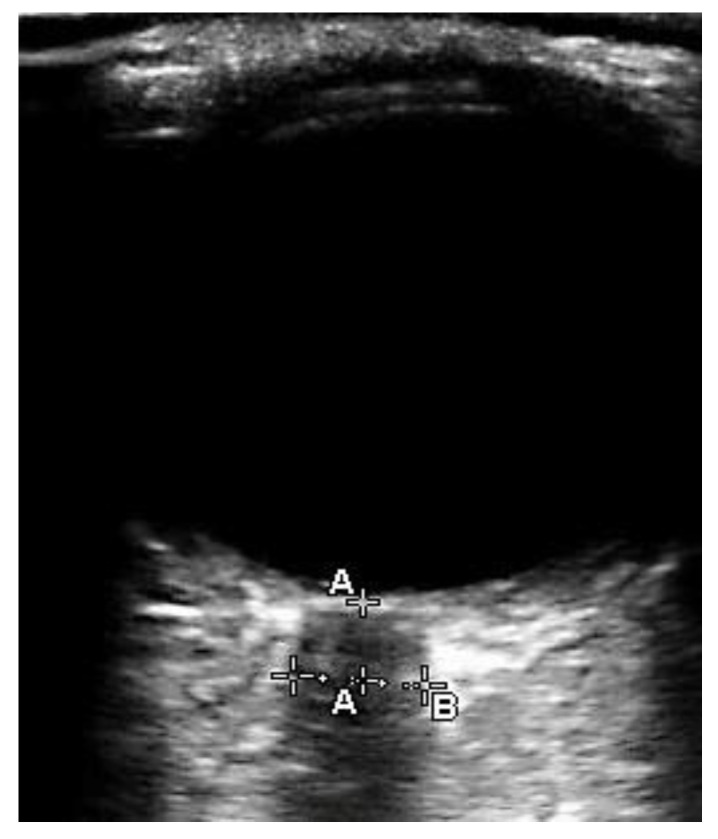
Ultrasonography image of optic nerve sheath. Optic nerve sheath diameter (**B**: 4.7 mm) is measured 3 mm behind the optic disc (**A**).

**Figure 2 jcm-11-02161-f002:**
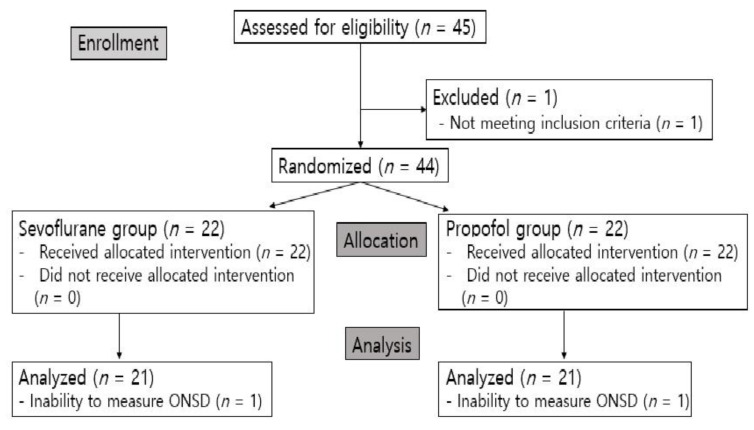
CONSORT flow gram.

**Table 1 jcm-11-02161-t001:** Demographic and perioperative data.

Variables	Sevoflurane Group (*n* = 21)	Propofol Group (*n* = 21)	*p*
Age (years)	44.4 ± 5.5	44.9 ± 6.8	0.805
Height (cm)	158.5 ± 4.7	158 ± 5.1	0.778
Weight (kg)	58.6 ± 6.5	61 ± 8.4	0.289
Body mass index (kg/m^2^)	23.2 ± 2.6	24.2 ± 3.3	0.282
Hypertension	3 (14.3)	1 (4.8)	0.606
Diabetes mellitus	1 (4.8)	1 (4.8)	1.000
Operation time (min)	140 (105–145)	130 (120–151)	0.791
Anesthesia time (min)	175 (145–195)	170 (155–191)	1.000
Crystalloid amount (mL)	1381 ± 278.6	1395.2 ± 327.5	0.880
Estimated blood loss (mL)	200 (200–300)	200 (200–300)	0.379

Data are shown as mean ± standard deviation or median (interquartile range) or number (%).

**Table 2 jcm-11-02161-t002:** Comparison of change in ONSD between two groups.

	Sevoflurane Group (*n* = 21)	Propofol Group (*n* = 21)	Difference in the Means	95% CI	*p*
T0 (mm)	4.54 ± 0.38	4.6 ± 0.26	−0.06	−0.27 to 0.14	0.530
T1 (mm)	4.94 ± 0.41	4.81 ± 0.27	0.13	−0.09 to 0.35	0.236
T2 (mm)	5.08 ± 0.47	4.85 ± 0.29	0.23	−0.02 to 0.47	0.070
T3 (mm)	5.18 ± 0.38	4.93 ± 0.31	0.25	0.03 to 0.47	0.025
T4 (mm)	4.59 ± 0.39	4.61 ± 0.26	−0.02	−0.22 to 0.19	0.872
Change in ONSD at T1 (mm)	0.41 ± 0.26	0.21 ± 0.13	0.19	0.06 to 0.32	0.005
Change in ONSD at T2 (mm)	0.54 ± 0.30	0.25 ± 0.09	0.29	0.15 to 0.43	<0.001
Change in ONSD at T3 (mm)	0.64 ± 0.23	0.33 ± 0.11	0.31	0.2 to 4.3	<0.001
Change in ONSD at T4 (mm)	0.05 ± 0.13	0.00 ± 0.07	0.05	−0.02 to 0.12	0.126
Maximum increase from baseline (%)	14.1 ± 5.08	7.17 ± 2.48	6.93	4.52 to 9.56	<0.001
Number					
Patients with ONSD over 5 mm at T0	2 (9.5)	1 (4.8)			0.549
Patients with ONSD over 5 mm at T1	11 (52.4)	8 (38.1)			0.352
Patients with ONSD over 5 mm at T2	15 (71.4)	11 (52.4)			0.204
Patients with ONSD over 5 mm at T3	15 (71.4)	11 (52.4)			0.204
Patients with ONSD over 5 mm at T4	4 (19)	1 (4.8)			0.153

Data are shown as mean ± standard deviation or number (%). T0 = 10 min after induction of anesthesia, T1 = 5 min after carbon dioxide pneumoperitoneum and steep Trendelenburg position, T2 = 20 min after carbon dioxide pneumoperitoneum and steep Trendelenburg position, T3 = 40 min after carbon dioxide pneumoperitoneum and steep Trendelenburg position, T4 = at skin closure after desufflation of pneumoperitoneum in the supine position. Change in ONSD at T1 = difference in ONSD between T1 and T0, Change in ONSD at T2 = difference in ONSD between T2 and T0, Change in ONSD at T3 = difference in ONSD between T3 and T0, Change in ONSD at T4 = difference in ONSD between T4 and T0, Maximum increase from baseline = difference between T3 and T0 in percentage. ONSD = optic nerve sheath diameter, CI = confidence interval.

**Table 3 jcm-11-02161-t003:** Comparison of respiratory and hemodynamic parameters between two groups.

	Sevoflurane Group (*n* = 21)	Propofol Group (*n* = 21)	*p*
EtCO_2_ (mmHg)			
T0	32 (32–33)	32 (32–33)	0.450
T1	34 (33–34)	34 (33–35)	0.949
T2	34 (34–35)	34 (33–35)	0.469
T3	34 (33–35)	34 (33–35)	0.601
T4	33 (32–34)	34 (32–34)	0.170
SBP (mmHg)			
T0	115.8 ± 18.1	115.7 ± 17.9	0.993
T1	124.4 ± 9.8	129.9 ± 11.5	0.105
T2	122.4 ± 12.1	128.2 ± 10.9	0.110
T3	120.0 ± 11.1	128.3 ± 8.2	0.009
T4	109.2 ± 11.6	115 ± 14.1	0.156
MBP (mmHg)			
T0	86.8 ± 15.1	84.9 ±13.8	0.664
T1	97.4 ± 8.2	98.8 ± 8.7	0.588
T2	95 ± 8.9	98.9 ± 8.5	0.165
T3	93.8 ± 8.4	97.7 ± 6.8	0.109
T4	82.6 ± 10.1	87.9 ± 11.8	0.128
DBP (mmHg)			
T0	67.3 ± 10.2	66.9 ± 12.2	0.913
T1	78.9 ± 7.7	79.9 ± 7.2	0.682
T2	77.1 ± 8.7	80.1 ± 8.0	0.259
T3	76.1 ± 7.9	78.8 ± 5.4	0.193
T4	64.4 ± 8.7	66.4 ± 9.4	0.468
Heart rate (beats/min)			
T0	77.2 ± 13.7	67.1 ± 10.9	0.012
T1	68.2 ± 10.3	62.8 ± 8.4	0.071
T2	69.4 ± 11.1	64.1 ± 9.5	0.108
T3	68.7 ± 9.8	63.3 ± 9.4	0.074
T4	64.4 ± 8.8	58.6 ± 7.8	0.028
SPO_2_ (%)			
T0	99.6 ± 0.6	99.4 ± 0.8	0.389
T1	99.8 ± 0.4	99.6 ± 0.6	0.246
T2	99.8 ± 0.4	99.6 ± 0.6	0.139
T3	99.8 ± 0.4	99.6 ± 0.6	0.229
T4	99.8 ± 0.4	99.9 ± 0.3	0.225
PPV (%)			
T0	5.1 ± 2.8	3.8 ± 2.6	0.114
T1	6.4 ± 3.4	6.2 ± 2.9	0.846
T2	6.0 ± 3.2	6.1 ± 2.3	0.871
T3	6 (4–7)	5 (4–7)	0.909
T4	6 (3–7)	3 (3–7)	0.291
PAP (cmH_2_O)			
T0	12 (10–13)	13 (12–15)	0.027
T1	22 ± 3.9	23.5 ± 4	0.220
T2	21.6 ± 3.8	23.2 ± 3.8	0.190
T3	21.4 ± 4.3	23.7 ± 3.4	0.061
T4	13.6 ± 2.6	14.5 ± 2.6	0.268

Data are shown as mean ± standard deviation or median (interquartile range). End-tidal carbon dioxide = EtCO_2_, systolic blood pressure = SBP, mean blood pressure = MBP, diastolic blood pressure = DBP, pulse oximetry oxygen saturation = SPO_2,_ pulse pressure variation = PPV, peak airway pressure = PAP.

## Data Availability

The data are presented within the article. Additional data are available on request from the corresponding author.

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
