# Peer review of "Comparison of the Effects of Propofol and Sevoflurane Anesthesia on Optic Nerve Sheath Diameter in Robot-Assisted Laparoscopic Gynecology Surgery: A Randomized Controlled Trial"

_jcm, 2022, doi:10.3390/jcm11082161_

Round 1
Reviewer 1 Report
I thank the Editor for giving me the opportunity to correct this valuable manuscript.
I thank the Authors for the interesting analysis carried out on a subject very dear to us clinicians: multi modal neuro monitoring. The Authors of “Comparison of the Effects of Propofol and Sevoflurane Anesthesia on Optic Nerve Sheath Diameter in Robot-assisted Laparoscopic Gynecology Surgery: A Randomized Controlled Trial”, have analyzed the effects on ONSD of two different anesthesia technique with an RCT, however, to be even more incisive the work, I would like to add some correct little ones, I hope useful.
1)Abstract: please, revise the Abstract following the revised version of the manuscript. In particular, the two sentences referring to PONV are correctly positioned at the end of the Results description. Please, also revise the sentence referred to “a preferable anesthetic method”
2)Introduction, line 31-33: even laparoscopic surgery requires a steep Trendelenburg position and carbon dioxide pneumoperitoneum
3)Materials and Methods, 2.2. Anesthetic and Surgical Techniques: please, two dosages of remifentanil used must be expressed with the same units of measure / modality, to make the method used understandable to the reader.
The authors describe in the same paragraph the use of noninvasive blood pressure measurement and then the cannulation of the radial artery for pulse pressure variation. Please standardize the message.
4)Materials and Methods, 2.3. Measurements: The Authors write that the anesthetist performing ultrasound "who did not know which group the patient was in", but then later describe that "During ONSD measurement, 95 the anesthetic vaporizer and target-controlled infusion pump were covered with opaque coverings. " Maybe, these two massages go together?
The Authors write that the vital signs were "measured" at 5 different times, but perhaps they meant collected. I think vital signs were measured continuously during anesthesia, right?
5)Materials and Methods, 2.4. Statistical Analysis: “All data were analyzed with SPSS software version 18.0 (IBM Corp., Armonk, NY, 114 USA).” Usually this sentence is at the end of the paragraph.
6)Results, line 136: “…statistically significantly…” two consecutive adverbs, they sound very bad. Please, revise.
7)Results, line 137-138: the Authors wrote “The groups had statistically significantly different systolic blood pressure 137 at T3 and peak airway pressures at T0.” but these results do not seem to transpire from table 3. Please, justify of remove
8)Figure 1: in the image it see very little and badly ONSD: I'm sure the Authors have produced and saved better images. It is important to always provide readers with credible references.
9)Table 2: what value exactly expresses "Change in ONSD at ...", is not explained in the caption. In particular, at which values does “Change in ONSD at T4” start?
Does "Maximum increase from baseline (%)" refer to T0 - T3? this is also not specified in the caption.
10)Table 3: SBP and PAP, should they contain significant data in T0 and in T3, as indicated in the results? Please, look at the remark at the number 7
11)Discussion, line 200-202: the Authors used the example of the duration of the surgery: the data cited seem to be very not significant. I don't think this example adds much to the message of the manuscript.
12)Discussion, line 218-220: the Authors wrote “Propofol anesthesia should be administered to women undergoing robotic surgery given its neuroprotective property and its reduced likelihood of causing PONV.” seems a conclusion that goes well beyond the aim of the present RCT, there are many high factors that must be considered, in the balance of advantages / disadvantages of the best anesthetic technique. I would remove this message from the Discussion and Conclusions, so as not to mislead the reader.
13)Discussion, line 221-222: I honestly do not think it is a limitation that all the ultrasound ONSD measurements have been performed by the same operator. Ultrosound is an operator-dependent procedure, so at least this possible bias is reduced. In addition to indicating the data on learning curves generically, it would be useful to indicate whether the operator used a standard bundle to minimize the error.
14)Conclusions: please, see observation in point 12
Please review the entire manuscript, also from a linguistic point of view. I hope that these little indications of mine, can help the publication of this manuscript.
Author Response
Response to Reviewer 1 Comments
Thank you very much for the thoughtful and thorough review.
Point 1: Abstract: please, revise the Abstract following the revised version of the manuscript. In particular, the two sentences referring to PONV are correctly positioned at the end of the Results description. Please, also revise the sentence referred to “a preferable anesthetic method”
Response 1: Thank you for your comment. In line with your recommendation mentioned in point12, we erased the last sentence of the abstract. Now, it is more focused on our original topic. Revised abstract reads as follows.
# Abstract
…. This study suggests that propofol anesthesia caused a lower increase in ONSD than sevoflurane anesthesia. This result suggests that propofol anesthesia is a preferable anesthetic method during robot-assisted laparoscopic gynecology surgery.
Point 2: Introduction, line 31-33: even laparoscopic surgery requires a steep Trendelenburg position and carbon dioxide pneumoperitoneum
Response 2: In my point of view, I feel many clinicians advocate steeper Trendelenburg position during robotic surgery (even to maximum degree) compared to conventional laparoscopic surgery. This is also mentioned by Ghomi A et al., while there is no consensus in advocating maximum Trendelenburg position during robotic surgeries [J Minim Invasive Gynecol. 2012 Jul-Aug;19(4):485-9.]. However, as your opinion, steep Trendelenburg position and pneumoperitoneum applies to both robotic and laparoscopic surgery. The harmful effects of Trendelenburg and pneumoperitoneum on the ONSD are the same for both robotic surgery and conventional laparoscopic surgery, so we revised the manuscript as follows.
# Introduction.
However, it requires patients to be placed in a steep Trendelenburg position and the induction of carbon dioxide pneumoperitoneum for surgical exposure, both of which increase intracranial pressure (ICP) during robotic surgeries likewise in conventional laparoscopic surgeries.
Point 3: Materials and Methods, 2.2. Anesthetic and Surgical Techniques: please, two dosages of remifentanil used must be expressed with the same units of measure / modality, to make the method used understandable to the reader.
Response 3: We used remifentanil using target controlled infusion pump for propofol group and conventional syringe pump was used for sevoflurane group. This is a common anesthetic method in our center. During the course of the study, we found the necessity of standardising the infusion method of remifentanil (unfortunately too late).
Following is the total amount of remifentanil used during anesthesia. Although there was no significant difference in the amount of remifentanil used, I don’t think this will give additional valuable information to the readers that I did not add this in Table.
|
|
Sevoflurane group |
Propofol group |
P-value |
|
Remifentanil (mcg) |
646.95 ± 261 |
691.1 ± 144 |
0.502 |
Point 3 continued: The authors describe in the same paragraph the use of noninvasive blood pressure measurement and then the cannulation of the radial artery for pulse pressure variation. Please standardize the message.
Response 3: Noninvasive blood pressure was measured once the patient arrived in the operating room. After the patient was induced anesthesia, continuous arterial pressure monitoring was used for analysis of the data. To prevent confusion of the readers, we revised as follows.
# 2.2 Anesthetic and Surgical Techniques
The radial artery was then cannulated for continuous arterial blood pressure monitoring and was recorded for analysis.
Point 4: Materials and Methods, 2.3. Measurements: The Authors write that the anesthetist performing ultrasound "who did not know which group the patient was in", but then later describe that "During ONSD measurement, 95 the anesthetic vaporizer and target-controlled infusion pump were covered with opaque coverings. " Maybe, these two massages go together?
Response 4: Yes. I agree with your opinion and revised the manuscript as follows.
# 2.3 Measurements
The ONSD was ultrasonographically measured with a 7.5 MHz linear probe by an experienced anesthesiologist (IJJ) who did not know which group the patient was in. The investigator had an extensive experience of ultrasonographic ONSD measurement of more than 100 scans. After 25 scans, a wide range of physicians can reliably measure the ONSD ultrasonographically [11,12]. During ONSD measurement, the anesthetic vaporizer and target-controlled infusion pump were covered with opaque coverings. Standardized criteria was used to optimize ONSD measurement [13]. The linear probe was placed taped to the patients’ closed eyelids with transparent tegaderm after ultrasound gel was applied. The vitreous body, optic disc, and hypoechoic optic nerve sheath were visualized by gently adjusting the probe angle (Fig. 2). By using electronic calipers, the ONSD was measured vertically 3 mm behind the optic disc in the sagittal and transverse planes in both eyes. We used the average of the four values for analysis. Each measurement was completed within one minute. During ONSD measurement, the anesthetic vaporizer and target-controlled infusion pump were covered with opaque coverings
Point 4 continued: The Authors write that the vital signs were "measured" at 5 different times, but perhaps they meant collected. I think vital signs were measured continuously during anesthesia, right?
Response 4: Yes. Vital signs were continuously measured and were collected at 5 different times. To clarify this, we revised the manuscript as follows.
# 2.3 Measurements
Hemodynamic and respiratory parameters, including end-tidal carbon dioxide, blood pressure, heart rate, pulse oximetry oxygen saturation, pulse pressure variation, and peak airway pressure were also continuously measured and collected at each of these five points in time.
Point 5: Materials and Methods, 2.4. Statistical Analysis: “All data were analyzed with SPSS software version 18.0 (IBM Corp., Armonk, NY, 114 USA).” Usually this sentence is at the end of the paragraph.
Response 5: Thank you for your comment. We revised as your recommendation.
Point 6: Results, line 136: “…statistically significantly…” two consecutive adverbs, they sound very bad. Please, revise.
Response 6: As your recommendation, we erased “significantly” from the sentence.
“…statistically significantly…” is repeated in other parts of the paper, we revised them also.
Point 7: Results, line 137-138: the Authors wrote “The groups had statistically significantly different systolic blood pressure at T3 and peak airway pressures at T0.” but these results do not seem to transpire from table 3. Please, justify or remove.
Response 7: Thank you for your recommendation. We suppose “systolic blood pressure at T3 and peak airway pressures at T0” is not something to have emphasis on, we erased this sentence from the result paragraph.
Point 8: in the image it see very little and badly ONSD: I'm sure the Authors have produced and saved better images. It is important to always provide readers with credible references.
Response 8: As your recommendation, we changed to a clear image.
Point 9: Table 2: what value exactly expresses "Change in ONSD at ...", is not explained in the caption. In particular, at which values does “Change in ONSD at T4” start? Does "Maximum increase from baseline (%)" refer to T0 - T3? this is also not specified in the caption.
Response 9: For better understading, we included the followings in the caption.
# Table 2 Caption.
Change in ONSD at T1 = difference of ONSD between T1 and T0, Change in ONSD at T2 = difference of ONSD between T2 and T0, Change in ONSD at T3 = difference of ONSD between T3 and T0, Change in ONSD at T4 = difference of ONSD between T4 and T0, Maximum increase from baseline = difference between T3 and T0 in percentage.
Point 10: Table 3: SBP and PAP, should they contain significant data in T0 and in T3, as indicated in the results? Please, look at the remark at the number 7
Response 10: We suppose “systolic blood pressure at T3 and peak airway pressures at T0” is not something to have emphasis on, we erased this sentence from the result paragraph.
Point 11: Discussion, line 200-202: the Authors used the example of the duration of the surgery: the data cited seem to be very not significant. I don't think this example adds much to the message of the manuscript.
Response 11: Yes, we agree with your opinion. We erased the sentence describing the difference of duration of surgery from the paragraph.
Point 12: Discussion, line 218-220: the Authors wrote “Propofol anesthesia should be administered to women undergoing robotic surgery given its neuroprotective property and its reduced likelihood of causing PONV.” seems a conclusion that goes well beyond the aim of the present RCT, there are many high factors that must be considered, in the balance of advantages / disadvantages of the best anesthetic technique. I would remove this message from the Discussion and Conclusions, so as not to mislead the reader.
Response 12: According to your recommendation, we erased the sentence from Discussion (#line 218-220) and revised the conclusion as follows.
# conclusion
The ONSD increased significantly in sevoflurane group compared to propofol group at 40 minutes with pneumoperitoneum in a Trendelenburg position. Propofol anesthesia produced smaller ONSD increases than sevoflurane anesthesia in women undergoing robot-assisted surgery. with pneumoperitoneum in a Trendelenburg position. Thus, propofol would be a better choice of anesthetic for robot-assisted gynecology surgery than sevoflurane.
Point 13: Discussion, line 221-222: I honestly do not think it is a limitation that all the ultrasound ONSD measurements have been performed by the same operator. Ultrosound is an operator-dependent procedure, so at least this possible bias is reduced. In addition to indicating the data on learning curves generically, it would be useful to indicate whether the operator used a standard bundle to minimize the error.
Response 13: We agree. We erased the first limitation and revised the # 2.3 measurements as follows.
# 2.3 Measurements
The ONSD was ultrasonographically measured with a 7.5 MHz linear probe by an experienced anesthesiologist (IJJ) who did not know which group the patient was in. The investigator had an extensive experience of ultrasonographic ONSD measurement of more than 100 scans. After 25 scans, a wide range of physicians can reliably measure the ONSD ultrasonographically [11,12]. During ONSD measurement, the anesthetic vaporizer and target-controlled infusion pump were covered with opaque coverings. Standardized criteria was used to optimize ONSD measurement [13]. The linear probe was placed taped to the patients’ closed eyelids with transparent tegaderm after ultrasound gel was applied. The vitreous body, optic disc, and hypoechoic optic nerve sheath were visualized by gently adjusting the probe angle (Fig. 2). By using electronic calipers, the ONSD was measured vertically 3 mm behind the optic disc in the sagittal and transverse planes in both eyes. We used the average of the four values for analysis. Each measurement was completed within one minute.
# Discussion - limitation
This study had three two major limitations. First, the ONSD was measured by a single investigator (IJJ). However, the investigator had an extensive experience of ultrasonographic ONSD measurement of more than 100 scans. After 25 scans, a wide range of physicians can reliably measure the ONSD ultrasonographically [30,31]. Standardized criteria were also used for optimizing ONSD measurement [32]. Second, First, the ONSD was monitored until 40 minutes after pneumoperitoneum was induced and the patient was put in a Trendelenburg position. ..
Point 14: Conclusions: please, see observation in point 12. Please review the entire manuscript, also from a linguistic point of view. I hope that these little indications of mine, can help the publication of this manuscript.
Response 14: We revised conclusion as mentioned in response 12.
Again, we thank you for your meticulous comments. I feel revised manuscript according to your recommendation is much more focused on the topic. If there is anything unclear in the text, please contact me anytime.
Reviewer 2 Report
Line 134, Table 2 An ONSD above 5 mm indicates an increase in intracranial pressure. It will be more informative if the number of patients over 5 mm is given in the form of n (%) in the time intervals examined. I think that whether this data makes a difference between the groups will enrich the findings of the study.
Table 3 should explain why the median (min-max) values are preferred over the mean standard deviation for the T0 value of PAP.
Yours Sincerely
Author Response
Response to Reviewer 2 Comments
Thank you very much for the thoughtful and thorough review.
Point 1: An ONSD above 5mm indicates an increase in intracranial pressure. It will be more informative if the number of patients over 5mm is given in the form of n(%) in the time intervals examined. I think that whether this data makes a difference between the groups will enrich the findings of the study.
Response 1: Thank you for your comment. As your recommendation, I suppose readers will want to know the number of patient over 5mm during measurement periods. We included the data in Table 2. This definitely enriches our findings.
Point 2: Table 3, T0: I should be explained why the mean standard deviation is not given for the T0 value.
Response 2: Thank you for your thoughtful comment. In Table 3, we described several values using median with interquartile range. Shapiro-Wilk test was done to test normality of all continuous data and we found the value at PAP at T0, EtCO2, … did not follow normal distribution. That way, we analyzed the data with Mann-Whitney U test and described the values using median (interquartile range) instead of mean (SD) on discussion with my statistician. I hope this has been explained. For anything you might feel insufficient, please let me know at any time.